# In Vitro and In Vivo Evaluation for Antioxidant and Anti-Diabetic Properties of *Cyperus rotundus* L. Kombucha

**DOI:** 10.3390/foods12224059

**Published:** 2023-11-08

**Authors:** Ananya Dechakhamphu, Nattapong Wongchum, Theeraphan Chumroenphat, Alongklod Tanomtong, Somchai Pinlaor, Sirithon Siriamornpun

**Affiliations:** 1Thai Traditional Medicine Program, Faculty of Thai Traditional and Alternative Medicine, Ubon Ratchathani Rajabhat University, Ubonratchathani 34000, Thailand; ananya.d@ubru.ac.th (A.D.); theeraphan.c@ubru.ac.th (T.C.); 2Aesthetic Sciences and Health Program, Faculty of Thai Traditional and Alternative Medicine, Ubon Ratchathani Rajabhat University, Ubonratchathani 34000, Thailand; 3Biology Program, Faculty of Science, Ubon Ratchathani Rajabhat University, Ubonratchathani 34000, Thailand; nattapong.w@ubru.ac.th; 4Biology Program, Faculty of Science, Khon Kaen University, Khon Kaen 40002, Thailand; alotan@kku.ac.th; 5Department of Parasitology, Faculty of Medicine, Khon Kaen University, Khon Kaen 40002, Thailand; psomec@kku.ac.th; 6Research Unit of Thai Food Innovation, Department of Food Technology and Nutrition, Mahasarakham University, Kantarawichai, Maha Sarakham 44150, Thailand

**Keywords:** functional food, health benefit, herbal medicine, health promotion

## Abstract

*Cyperus rotundus* L. exhibits promising potential for the development of functional foods due to its documented pharmacological and biological activities. This study investigated the antioxidant and anti-diabetic properties of *C. rotundus* kombucha. The results demonstrated potent antioxidant activity with an IC_50_ value of 76.7 ± 9.6 µL/mL for the DPPH assay and 314.2 ± 16.9 µL/mL for the ABTS assay. Additionally, the kombucha demonstrated alpha-glucosidase inhibitory with an IC_50_ value of 142.7 ± 5.2 µL/mL. This in vitro antioxidant potential was further validated in vivo using *Drosophila*. *Drosophila* fed a high-sugar diet and supplemented with pure kombucha revealed significant increases in DPPH and ABTS free radical scavenging activity. *Drosophila* on a high-sugar diet supplemented with varying kombucha concentrations manifested enhanced resistance to oxidative stresses induced by H_2_O_2_ and paraquat. Concurrently, there was a notable decline in lipid peroxidation levels. Additionally, significant upregulations in CAT, SOD1, and SOD2 activities were observed when the high-sugar diet was supplemented with kombucha. Furthermore, in vivo assessments using *Drosophila* demonstrated significant reductions in alpha-glucosidase activity when fed with kombucha (reduced by 34.04%, 13.79%, and 11.60% when treated with 100%, 40%, and 10% kombucha, respectively). A comprehensive GC-MS and HPLC analysis of *C. rotundus* kombucha detected the presence of antioxidative and anti-glucosidase compounds. In conclusion, *C. rotundus* kombucha exhibits considerable antioxidant and anti-diabetic properties, demonstrating its potential as a beneficial beverage for health promotion.

## 1. Introduction

*Cyperus rotundus* L. from the Cyperaceae family, also known as purple nutsedge, is documented as a longevity remedy in ancient Thai manuscripts. Historically, this herb has been employed in traditional medicinal practices across China, India, Africa, Japan, and Arab nations to treat various ailments [1]. Numerous studies have provided evidence that the tubers and rhizomes of this particular plant have a wide range of therapeutic properties. These include antioxidant [2,3,4], anti-diabetic [5,6], and anti-obesity effects [7]. They also have antiallergic [8], antimicrobial [9], and antidiarrheal capabilities [10]. Moreover, they offer cardioprotective [11], gastroprotective [12], and hepatoprotective benefits [13]. The plant also exhibits immunomodulatory [14], neuroprotective [15,16], and anticarcinogenic properties [17]. Additionally, it has been found to have antiarthritic [18,19,20], anti-inflammatory, anti-uropathogenic [21], anticonvulsant [22], and antidepressant effects [23]. Recently reports have revealed that the hydroethanolic extract of *C. rotundus* enhances sexual behavior and fitness in *Drosophila* [24]. The supplementation of *Drosophila* with this extract resulted in an increase in lifespan and a decrease in oxidative stress-induced mortality [25]. In a *Drosophila* obesity model, supplementation with *C. rotundus* extract exhibited to extend lifespan in the presence of high-fat diet-induced mortality [26]. Given its demonstrated biological and pharmacological properties, this plant’s rhizomes and tubers have the potential to be used in the development of functional foods and beverages. These products may offer viable alternatives for disease prevention and therapeutic interventions.

The inhibition of alpha-glucosidase is critical for the effective management of type-2 diabetes, which is distinguished by hyperglycemia, resistance to insulin, and insufficient insulin secretion [27]. By decelerating the conversion of carbohydrates into simple sugars, alpha-glucosidase suppressants can reduce post-meal glucose increases, facilitating in maintaining of steady blood sugar levels and functioning as an effective diabetes treatment. In addition to their utility in the treatment of diabetes, alpha-glucosidase inhibitors have attracted attention for their therapeutic potential in treating a variety of diseases [28]. Some findings indicate that a delayed carbohydrate digestion process might increase the feeling of fullness, potentially reducing overall calorie consumption and nurturing weight reduction [29]. Currently, the majority of anti-glucosidase action research is conducted in in vitro settings, which may not replicate the complexity of living conditions. Animal research should be intensive in order to discover and validate promising drug candidates for the management of diabetes.

In scientific literature, kombucha is acknowledged for its putative health benefits, which range from improved digestive and immune function to cholesterol level modulation and detoxification processes [30,31]. Although empirical research addressing these assertions is still in its early stages, the beverage’s rising popularity is influenced by the global trend toward natural and probiotics-enriched foods and beverages. Numerous in vitro studies have investigated the bioactivities of kombucha and its constituents, but the number of published in vivo studies is significantly less. To obtain a comprehensive understanding, it is essential to expand in vivo studies, conduct detailed bio-accessibility and bioavailability assessments, and precisely identify bioactive compounds. These initiatives will facilitate a thorough investigation of distinctive molecular functions.

*Drosophila melanogaster* or fruit fly is recognized as a model organism for investigating metabolic disorders. This organism has contributed to the development of diabetes models that replicate the complexities of type-2 diabetes in humans. A technique for inducing T2D symptoms in *Drosophila* via high-sugar diets has been developed [32]. Comparable to human physiology, *Drosophila* contains insulin-producing cells, insulin-like molecules, and an insulin-receptive mechanism [33]. In both the larval and adult stages of *Drosophila*, characteristics similar to insulin resistance, such as metabolic disorders and disrupted insulin signaling, can be replicated [34]. The established effects of high-sugar diets on *Drosophila* include high blood sugar levels, insulin resistance, increased fat accumulation, and decreased lifespan [35]. In addition, researchers have investigated the effect of botanical extracts on preventing metabolic disruptions in *Drosophila* caused by a high sucrose intake. Additionally, fruit flies contain the glucosidase enzyme, which is essential for sugar adaptation [36]. Therefore, *Drosophila* is a suitable model for evaluating the alpha-glucosidase-inhibitory properties of specific compounds.

In this study, we investigated the antioxidant and anti-diabetic properties of *C. rotundus* kombucha both in vitro and in vivo. Additionally, we analyzed its phytochemical composition. Our findings serve to bridge the gap between in vitro observations and in vivo conditions, potentially emphasizing the prospective role of *C. rotundus* kombucha as a functional beverage for disease prevention or intervention.

## 2. Materials and Methods

### 2.1. Chemicals

a,a-Diphenyl-b-picrylhydrazyl (DPPH), 2,20-azino-bis-(3-ethylbenzothiazoline- 6-sulfonic acid) diammonium salt (ABTS), sodium phosphate monobasic, α-glucosidase, hydrogen peroxide, 1,1′-dimethyl-4,4′-bi-pyridinium dichloride (paraquat dichloride hydrate), CAT assay kit, SOD assay kit, and sodium phosphate dibasic were purchased from the Sigma Chemical Co. (St. Louis, MO, USA). The SCOBY agent was purchased from Master Lab Company (Ubonratchathani, Thailand).

### 2.2. Plant Materials and Preparation of Kombucha

*C. rotundus* rhizomes and tubers were collected from crop fields in the Thai province of Singburi in April 2022. The Nattapong 01 voucher specimen was preserved at the Mahasarakham University Herbarium, with a collection date of 21 April 2022. The plant samples were washed and then dried for two days in a 50 °C hot air oven. After drying, the materials were pulverized to a fine powder. *C. rotundus* kombucha was prepared by boiling with 1% (*w*/*v*) *C. rotundus* powder for 10 min. After boiling, it was filtered through cheesecloth at a temperature of 95 °C. Subsequently, 5% sugar (*w*/*v*) was added, and the mixture was boiled for an additional 5 min. The boiled solution was then transferred to a sterile container. Once boiled solution was cooled, a 10% SCOBY (*w*/*v*) was added. The container was covered with cheesecloth and left to ferment at room temperature for 10 days.

### 2.3. GC-MS Analysis of C. rotundus Kombucha

Phytochemical analysis of the *C. rotundus* kombucha was performed utilizing a GC-MS apparatus (7890B GC/5977B MSD, Agilent Technologies, Santa Clara, CA, USA) equipped with a 30 m HP-5MS capillary column, featuring a 250 mm diameter and a phase thickness of 0.25 mm. The kombucha sample was proportionally diluted with ethanol and introduced in a splitless mode. The oven started at 50 °C, maintained for 0.5 min, subsequently elevated to 250 °C at a rate of 10 °C per minute, and was sustained at this temperature for an additional 5 min. Helium (99.999%) served as the carrier gas with a regulated flow of 1.2 mL/min. The MS source temperature remained fixed at 230 °C, with a scan mode analysis spanning a mass range from 40 to 500. The compounds were identified using the NIST Mass Spectral Database.

### 2.4. HPLC Analysis of C. rotundus Kombucha

The evaluation of phenolic acids and flavonoids was conducted using high-performance liquid chromatography (HPLC) from Agilent Technology, Santa Clara, CA, USA. The apparatus consisted of a protective guard column and an InertSustain^®^ C18 column (250 mm × 4.6 mm i.d., 5 m; sourced from GL Sciences Inc., Tokyo, Japan). The mobile phase consisted of acetonitrile and 0.1% acetic acid in water, using following gradient: 0–5 min: 15% acetonitrile; 5–10 min: 25% acetonitrile; 10–20 min: 15% acetonitrile. The flow rate was 0.5 mL/min. At wavelengths of 280 and 320 nm, a diode array detector was employed to detect phenolic acids and flavonoids. The use of external standards facilitated the identification and comparison of individual phenolic acids and flavonoids within the samples.

### 2.5. In Vitro Antioxidant Activity

#### 2.5.1. Scavenging DPPH Free Radical Activity Assay

The inhibitory effect on DPPH free radical activity of kombucha was evaluated by mixing 180 μL of kombucha solution with 20 μL of a 1 mM DPPH solution (solubilized in methanol). Reaction mixtures were incubated for 30 min at room temperature under no light. The absorbance at a wavelength of 517 nm was measured using an ELISA reader (Biochrome, UK) to determine the inhibitory effect. Each sample was examined three replicates. The inhibitory activity was calculated using the following formula:Inhibition (%) = ((Abs of control − Abs of sample)/(Abs of control)) × 100
where “Abs of the control” refers to the DPPH solution’s absorbance value without the sample.

The 50% inhibitory concentration (IC_50_) was established from the regression equation derived from plotting the percentage of inhibition against kombucha concentrations (0–800 µL/mL, diluted with sterile water).

#### 2.5.2. Scavenging ABTS Free Radical Activity Assay

The inhibitory effect of kombucha on ABTS free radical activity was assessed by mixing 20 µL of the kombucha solution with 180 µL of the ABTS^•^ solution. The latter was prepared by combining 10 mL of 7 mM ABTS (in ultrapure water) with 10 mL of 2.45 mM ammonium persulphate and then incubating it overnight (12–16 h) at room temperature in the dark. The concentration of the ABTS radical (ABTS^•^) stock solution was verified at 734 nm. An ABTS^•^ solution was prepared to achieve an absorption of approximately 0.700 at 734 nm. Reaction mixtures were incubated for 10 min at room temperature in the dark. The absorbance at a wavelength of 734 nm was measured using an ELISA reader (Biochrome, UK) to determine the inhibitory effect. Each sample was examined three replicates. The inhibitory activity was calculated using the following formula:Inhibition (%) = ((Abs of control − Abs of sample)/(Abs of control)) × 100
where “Abs of the control” refers to the ABTS^•^ solution’s absorbance value without the sample.

The 50% inhibitory concentration (IC_50_) was established from the regression equation derived from plotting the percentage of inhibition against kombucha concentrations (0–800 µL/mL, diluted with sterile water).

### 2.6. In Vitro Anti-Alpha Glucosidase Activity

The inhibitory activity of the extracts against α-glucosidase was evaluated using a method described by Pistia-Brueggeman and Hollingsworth [37], with slight modifications. Briefly, 50 µL of α-glucosidase solution (0.5 U/mL in 50 mM potassium phosphate buffer, pH 7.4) was mixed with 50 µL of either the kombucha solution or acarbose and incubated at room temperature for 10 min. Subsequently, 50 µL of 5 mM p-nitrophenyl-α-D-glucopyranoside (dissolved in 50 mM potassium phosphate buffer, pH 7.4) was added and the mixture was further incubated for 10 min at room temperature. The absorbance was then measured at 405 nm using a spectrophotometer (Biochrom, Cambridge, UK). The 50% inhibitory concentration (IC_50_) was established from the regression equation derived from plotting the percentage of inhibition against kombucha concentrations (0, 100, 200, 400, 600, and 800 uL/mL, diluted with sterile water).

### 2.7. Drosophila Strain, Culture Conditions, and Experimental Design

The wild-type *D. melanogaster* Oregon-R-C strain was donated by the Department of Biology at Khon Kaen University. The flies were housed in a wheat cream medium enriched with yeast powder at 25.1 ± 0.2 °C, 70–80% relative humidity, and subjected to a 12:12 light/dark cycle in a laboratory setting. Every two days, surviving flies were relocated to vials with fresh food. 

For the preparation of the high-sugar diet (HSD), 30 g of sucrose was dissolved in 100 mL of sterile water. In the case of the HSD + pure *C. rotundus* kombucha diet, 30 g of sucrose was dissolved in 100 mL of undiluted kombucha. For the HSD + 400 µL/mL (40%) and HSD + 100 µL/mL (10%) kombucha diets, kombucha was initially diluted to concentrations of 400 and 100 µL/mL, respectively, with sterile water. Thereafter, 30 g of sucrose was incorporated into each 100 mL of the diluted solutions.

Experiments were conducted using 5–7-day-old female flies, which were divided into four groups: HSD, HSD+ pure kombucha, HSD + 400 (40%) µL/mL kombucha, and HSD + 100 (10%) µL/mL kombucha. Each group consisted of 100 flies, with 20 flies per vial. The diets for each group were poured on the cotton sheet that placed in the bottom of the test vial (2.0 × 9.5 cm). Flies were deprived for two hours in an empty vial containing cotton cloth saturated with sterile water prior to treatment. After being starved for two hours, the flies were transferred to the test and fed for 4 h. The flies were transferred to an empty vial following the feeding period. Following euthanasia with 5% CO_2_ fumigation, the flies were chilled for ten minutes at −20 °C. The euthanized flies were subsequently transferred to a 1.5-mL microcentrifuge tube. The fly samples were washed three times with 1.0 mL of 50 mM potassium phosphate buffer. The samples were then homogenized using a pestle in 1.0 mL of the same buffer. The homogenate was subjected to a 10-min centrifugation at 10,000 rpm and 4 °C in order to remove fly debris. The supernatant was transferred to new 1.5 mL microcentrifuge tubes and kept on ice during preparation for measurement of antioxidant activity (DPPH and ABTS assays, CAT and SOD activities, and lipid peroxidation assay), α-glucosidase activity, and protein content. 

The research protocol received approval from the Ethics Committee of Ubon Ratchathani Rajabhat University (Ethical Clearance No. AN63008).

### 2.8. In Vivo Antioxidant Activity

#### 2.8.1. Scavenging DPPH Free Radical Activity Assay

The inhibitory effect on DPPH free radical activity in *Drosophila* homogenate was evaluated by mixing 180 μL of the supernatant with 20 μL of a 1 mM DPPH solution (solubilized in methanol). Reaction mixtures were incubated for 30 min at room temperature under no light. The absorbance at a wavelength of 517 nm was measured using an ELISA reader (Biochrome, UK) to determine the inhibitory effect. Each sample was examined three replicates. The inhibitory activity was calculated using the following formula:Inhibition (%) = ((Abs of control − Abs of sample)/(Abs of control)) × 100
where “Abs of the control” refers to the DPPH solution’s absorbance value without the sample.

The inhibitory activity was normalized to the protein content in the supernatant.

#### 2.8.2. Scavenging ABTS Free Radical Activity Assay

The inhibitory effect on ABTS free radical activity in *Drosophila* homogenate was evaluated by mixing 20 μL of the supernatant with 180 μL of ABTS^•^ solution. Reaction mixtures were incubated for 10 min at room temperature in the dark. The absorbance at a wavelength of 734 nm was measured using an ELISA reader (Biochrome, UK) to determine the inhibitory effect. Each sample was examined three replicates. The inhibitory activity was calculated using the following formula:Inhibition (%) = ((Abs of control − Abs of sample)/(Abs of control)) × 100
where “Abs of the control” refers to the ABTS^•^ solution’s absorbance value without the sample.

The inhibitory activity was normalized to the protein content in the supernatant.

#### 2.8.3. SOD Activity

SOD activity was determined using a Sigma-Aldrich reagent in accordance with the manufacturer’s instructions and the sample preparation procedure. Fruit flies (*n* = 100, 20/vial) were ground in 1 mL of cold buffer and then centrifuged (1500× *g* for 5 min at 4 °C) to remove any remaining material. The supernatant (900 µL) underwent another centrifugation (10,000× *g* for 15 min at 4 °C). This supernatant was then used to measure the activity of SOD1. To evaluate SOD2 activity, the resultant pellet was redissolved in 0.5 mL of cold buffer. Each test was performed three times.

#### 2.8.4. CAT Activity

The activity of CAT was evaluated using a kit from Sigma-Aldrich that measures the remaining hydrogen peroxide after the CAT reaction. In a summarized process, flies (*n* = 100, 20/vial) were ground in 1 mL of cold enzyme dilution buffer and then centrifuged (1500× *g* for 5 min at 4 °C) to eliminate any waste. A portion of the supernatant (500 µL) was then diluted in 20 mL of the assay buffer. It was combined with 25 µL of hydrogen peroxide solution and, after 1 min, the reaction was halted using 900 µL of 15 mM sodium azide. Subsequently, a 10 µL sample was introduced to a color reagent, which consisted of 0.25 mM 4-aminoantipyrine, 2 mM 3,5-dichloro-2-hydroxybenzenesulfinic acid, and peroxidase (0.5–1.5 U/mg). After incubating for 15 min at ambient temperature, the absorbance was recorded at 520 nm using an instrument supplied by UK-based Biochrome. This assay was repeated three times for accuracy.

#### 2.8.5. Measurement of Lipid Peroxidation

Lipid peroxidation was assessed using a modified method from Zeb and Ullah [38], employing the TBAR technique and taking malondialdehyde as a reference. In a concise process, a 500 µL portion of the fly homogenate was combined with 10 mL of TBA reagent. This mixture was then heated for 15 min, allowed to cool, and centrifuged at 3000 rpm for 10 min to remove any sediment. The absorbance of the remaining supernatant was recorded at 532 nm using a Biochrome instrument from the UK. The concentration of MDA was calculated based on a standard curve derived from 1,1,3,3-tetramethoxypropane.

#### 2.8.6. H_2_O_2_ and Paraquat Challenge Tests

In order to evaluate resistance to hydrogen peroxide and paraquat (superoxide anion), distinct sets of flies were utilized. Female flies, aged 5–7 days, were starved for 2 h before being transferred to containers with filter paper saturated with either a 20 mM paraquat solution or a 10% hydrogen peroxide solution. These solutions were either unaltered or supplemented with *C. rotundus* kombucha at concentrations of pure, 400 µL/mL, or 100 µL/mL, and each was prepared using a 6% sucrose solution. The number of deceased flies was noted at four-hour intervals until no flies remained alive.

### 2.9. In Vivo Anti-Alpha Glucosidase Activity

The glucosidase activity in *Drosophila* was evaluated by combining 50 μL of supernatant with 50 μL of 50 mM potassium phosphate buffer and 50 μL of 5 mM p-nitrophenyl-α-D-glucopyranoside (dissolved in 50 mM potassium phosphate buffer). The reaction mixtures were incubated at ambient temperature for 30 min. The glucosidase activity was quantified at a wavelength of 405 nm using an ELISA reader (Biochrome, UK). The standard curve for alpha-glucosidase was established by plotting the optical density at 405 nm against enzyme concentrations ranging from 0.03–0.5 U/mL. Each specimen was examined in triplicate. The activities of the enzymes were adjusted to the protein concentration in the supernatant.

### 2.10. Protein Content Assay

The protein content of the homogenate derived from the entire body was determined using the Bradford (1976) method and Merck’s Bradford reagent according to the manufacturer’s instructions. Concentrations were determined in mg/mL using bovine serum albumin as the standard.

### 2.11. Statistical Analysis

The data were statistically analyzed using SPSS 20.0 (SPSS Inc., Chicago, IL, USA). One-way ANOVA was used for mean comparisons. Kaplan–Meier analysis was used to determine the difference in survival curves. Statistical significance was defined as a *p* < 0.05. The significance levels for each experiment were denoted by the symbols * for *p* < 0.05 and ** for *p* < 0.01.

## 3. Results

### 3.1. Chemical Profiles of C. rotundus Kombucha

Table 1 shows the GC-MS-identified compounds of *C. rotundus* kombucha. It has been documented that the following compounds in kombucha exhibit biological activity: 3,5-dimethylpyrazole, 4H-pyran-4-one, 2,3-dihydro-3,5-dihydroxy-6-methyl, 2-(4-methyl-1H-1,2,3-triazol-1-yl)ethan-1-amine, 5-hydroxymethylfurfural, 10-azido-1-decanethiol, 3-deoxy-D-mannoic lactone, 1-.beta.-d-ribofuranosyl-3-[5-tetraazolyl]-1,2,4-triazole, Nona-2,3-dienoic acid, ethyl ester, and stigmasterol.

The phytochemical compounds of *C. rotundus* kombucha, analyzed using HPLC, are presented in Figure 1. The concentrations of chlorogenic acid and epicatechin were quantified using a standard curve. The range for chlorogenic acid was 3.13–50.00 µg/mL, described by the equation Y = 273,991x + 788,198. The range for epicatechin was 1.56–50.00 µg/mL, given by the equation Y = 166,769x + 560,043. As demonstrated in Figure 1, chlorogenic acid and epicatechin were detected at the concentration of 6.04 and 1.07 µg/g, respectively.

### 3.2. Effects of C. rotundus Kombucha on In Vitro and In Vivo Antioxidant Activity

The antioxidant properties of *C. rotundus* kombucha were evaluated using two distinct assays: DPPH and ABTS. In the evaluation, the kombucha solution produced by *C. rotundus* exhibited significant antioxidant properties. For the DPPH assay, the IC_50_ was calculated to be 76.7 ± 9.6 µL/mL (Figure 2A). The IC_50_ value for the kombucha solution in the ABTS assay was 314.2 ± 16.9 µL/mL (Figure 2B). The antioxidant findings suggest that *C. rotundus* kombucha has significant antioxidant potential, as evidenced by its significant inhibitory activity at low concentrations. Consequently, consuming *C. rotundus* kombucha may provide the body with a beneficial source of antioxidants.

The antioxidant potential of *C. rotundus* kombucha, as observed in vitro, was further confirmed in vivo using *Drosophila* as a test model, employing DPPH and ABTS assays. As shown in Figure 2C, *Drosophila* consuming a high-sugar diet supplemented with 100% kombucha exhibited a 45.44% increase in DPPH free radical scavenging activity relative to those consuming a high-sugar diet alone (*p* < 0.01). When the same diet was supplemented with 40% or 10% *C. rotundus* kombucha, the scavenging efficiency increased by 36.35% and 30.74%, respectively, compared to the HSD group (*p* < 0.05). Similar to the DPPH results, *Drosophila* on a high-sugar diet supplemented with 100% kombucha demonstrated a 21.22% increase in the ABTS free radical scavenging capacity compared to the HSD group (*p* < 0.01). Adding 40% or 10% *C. rotundus* kombucha to the diet increased scavenging rates by 13.52% and 8.39%, respectively, compared to the HSD group (*p* < 0.05) (Figure 2D). The findings of this study indicate that *C. rotundus* kombucha has significant antioxidant properties when consumed. This suggests that *C. rotundus* kombucha may be a beneficial beverage for reducing oxidative stress caused by free radicals.

Figure 3A illustrates the effect of *C. rotundus* kombucha on the resistance of *Drosophila* on a high-sugar diet to H_2_O_2_-induced oxidative stress. When compared to the *Drosophila* group on a high-sugar diet, those fed with pure kombucha on a high-sugar diet experienced a rise in their maximum and 50% survival rates by 13.23% and 12.52%, respectively *(p* < 0.05).

Figure 3B illustrates the effect of *C. rotundus* kombucha on the resistance of *Drosophila* fed a high-sugar diet to oxidative stress induced by paraquat. Compared to the group on a high-sugar diet alone, the *Drosophila* administered pure kombucha in addition to a high-sugar diet exhibited increases in their maximum and 50% survival rates by 21.74% and 23.67%, respectively (*p* < 0.01).

In this study, we evaluated the overall body lipid peroxidation (LPO) level to validate the antioxidant effects. As depicted in Figure 3C, *Drosophila* that consumed a high-sugar diet supplemented with pure kombucha demonstrated a notable reduction in LPO levels in comparison to the control group (*p* < 0.05).

The results shown in Figure 4A,B indicate that when the high-sugar diet was enriched with pure kombucha, there was a significant increase in CAT, SOD1, and SOD2 activities, compared to *Drosophila* that consumed only a high-sugar diet (*p* < 0.05).

### 3.3. Effects of C. rotundus Kombucha on In Vitro and In Vivo Alpha-Glucosidase Inhibitory Activity

The alpha-glucosidase inhibitory activity of the *C. rotundus* kombucha is shown in Figure 5A. The kombucha solution produced by *C. rotundus* showed inhibition potential with IC_50_ values of 142.7 ± 5.2 µL/mL. This result suggests that *C. rotundus* kombucha may function as an anti-diabetic beverage by inhibiting sugar absorption and, consequently, regulating blood glucose levels, thereby potentially preventing insulin resistance.

The in vitro observation of the inhibitory effect of *C. rotundus* kombucha on alpha-glucosidase activity was confirmed in vivo using a *Drosophila* model. As shown in Figure 5B, *Drosophila* fed a high-sugar diet supplemented with 100% kombucha demonstrated a 34.04% decrease in glucosidase activity compared to those fed a high-sugar diet alone (*p* < 0.001). When the high-sugar diet was supplemented with either 40% or 10% *C. rotundus* kombucha, glucosidase activity decreased by 13.79% (*p* < 0.05) and 11.60% (*p* < 0.05), respectively, compared to *Drosophila* fed only the high-sugar diet. These results demonstrate in vitro and in vivo that *C. rotundus* kombucha significantly reduces glucosidase activity, indicating a possible anti-diabetic effect.

## 4. Discussion

*C. rotundus* is a plant that has been used as medicine and food to promote health for a long time. Biological and pharmacological studies have been conducted both in vitro and in animals that demonstrate the beneficial effects of this plant. Therefore, the application of this plant in developing functional food to maintain health or prevent disease is very attractive. The purpose of this study was to process rhizomes and tubers into kombucha, a beverage that recently gained increasing recognition. To determine its health benefits, we investigated the antioxidant and anti-diabetic properties of *C. rotundus* kombucha both in vitro and in vivo. Our results revealed that *C. rotundus* kombucha exhibits potent antioxidant activity, particularly in scavenging DPPH and ABTS free radicals. Our findings suggest that *Drosophila* treated with kombucha are more resistant to oxidative stress induced by hydrogen peroxide and paraquat. This highlights the potent antioxidant properties of *C. rotundus* kombucha and indicates at the mechanism underlying its action.

Maintaining an appropriate balance between ROS production and elimination by various enzymatic and non-enzymatic agents is essential for cellular survival. In our research, we found that *Drosophila* treated with *C. rotundus* kombucha on a high-sugar diet exhibited enhanced activity of antioxidant enzymes, including SOD1, SOD2, and CAT, in comparison to flies on a high-sugar diet alone. This result is consistent with the findings of Wongchum et al. [26], who reported that the hydroethanolic extract from *C. rotundus* rhizome reduced paraquat-induced oxidative stress and increased the activity of antioxidant enzymes. Based on the chemical constituents found in the *C. rotundus* kombucha, Li et al. [59] showed that catechins from green tea can enhance the activity of antioxidant enzymes in *Drosophila*. The antioxidant capabilities of extracts might be attributed to the presence of phenols, flavonoids, and various other chemical constituents. GC-MS and HPLC analysis detected antioxidant-containing compounds, including 2-(4-methyl-1H-1,2,3-triazol-1-yl)ethan-1-amine [43], 4H-pyran-4-one, 2,3-dihydro-3,5-dihydroxy-6-methyl [42], 3-deoxy-D-mannoic lactone [47], stigmasterol [55], chlorogenic acid [60], and epicatechin [61]. The results of this study, combined with previous research, strongly suggest that the antioxidant potential of *C. rotundus* kombucha is linked to its antioxidant capacity and the activation of antioxidant enzymes.

A significant risk factor for the development of type-2 diabetes, which is characterized by elevated blood glucose levels, insulin resistance, and insufficient insulin secretion, is a high sugar intake. Reducing the activity of alpha-glucosidase is essential for managing and treating type-2 diabetes [27]. Similar to mammalian systems, *Drosophila* is a valuable model for investigating the anti-diabetic effects of natural substances [32]. Research has shown that a high-sugar diet leads to elevated blood sugar, insulin resistance, increased fat storage, and reduced life expectancy in *Drosophila* [35]. In addition, studies have revealed that fruit flies contain the glucosidase enzyme, which is essential for sugar absorption. In this study, we found that *C. rotundus* kombucha inhibited in vitro alpha-glucosidase activity with an IC_50_ of 142.7 ± 5.2 µL/mL. Compared to other plants, the review by Kashtoh and Baek [62] reported that the IC_50_ values for glucosidase activity varied from 0.53 ± 0.014 µg/mL to 1.873 ± 0.421 mg/mL. This suggests that *C. rotundus* kombucha is among the potent inhibitors. Our in vitro findings were verified in *Drosophila*; we observed a significant decrease in glucosidase activity in *Drosophila* fed a high-sugar diet supplemented with kombucha compared to those fed only a high-sugar diet. According to GC-MS and HPLC analysis, kombucha contains phytochemical compounds with anti-glucosidase properties. The evidence indicates that 3-deoxy-D-mannoic lactone derived from *Distichochlamys citrea* effectively inhibits α-glucosidase activity both in vitro and in silico [47]. Studies have shown that plant-derived stigmasterol can inhibit alpha-glucosidase activity [63,64,65]. Wang et al. [66] reported that chlorogenic acid inhibited alpha-glucosidase activity, similar to acarbose. Therefore, based on our findings and prior research, we suggest that *C. rotundus* kombucha might act as a preventative measure against diabetes by limiting dietary sugar absorption. Nonetheless, further studies are essential to comprehensively understand the underlying mechanisms.

## 5. Conclusions

Our research highlights the variety of health benefits associated with *C. rotundus*, which has been utilized historically for medicinal and dietary intentions. The processing of rhizomes and tubers of *C. rotundus* into kombucha, an increasingly popular health beverage, showed significant antioxidant properties, specifically in the inhibition of DPPH and ABTS free radicals. Furthermore, our study provides evidence that *C. rotundus* kombucha has been shown to enhance the activity of antioxidant enzymes (SOD1, SOD2, and CAT) in *Drosophila* in response to oxidative stress; this result is consistent with findings from previous research. The analysis of its components showed a diverse collection of compounds abundant in antioxidants. We observed that kombucha inhibited alpha-glucosidase activity in the context of type-2 diabetes, indicating that it may have the ability to reduce dietary sugar absorption. In conclusion, our study provides preliminary evidence supporting the potential of *C. rotundus* kombucha as an herbal remedy for enhancing antioxidant levels and preventing type-2 diabetes. In accordance with contemporary health demands and trends, we recommend further development of kombucha as a functional beverage intended to promote overall health and reduce the risk of chronic metabolic diseases.

## Figures and Tables

**Figure 1 foods-12-04059-f001:**
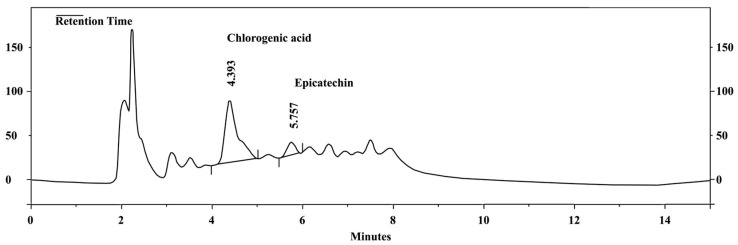
The HPLC chromatograms of *C. rotundus* kombucha reveal the presence of chlorogenic acid and epicatechin.

**Figure 2 foods-12-04059-f002:**
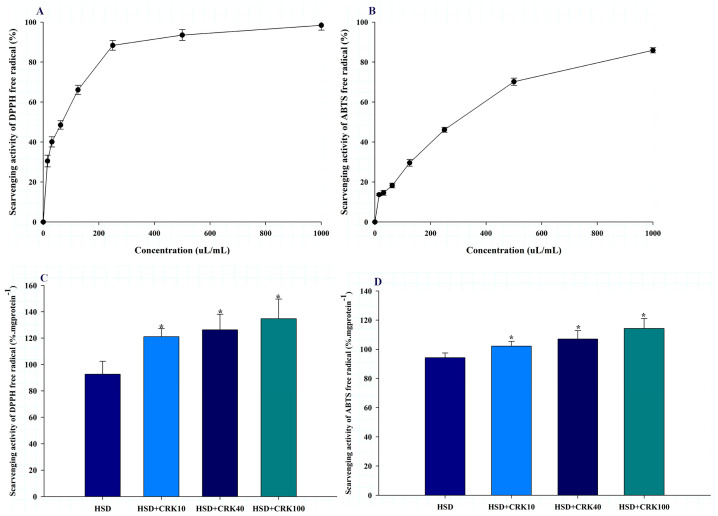
The effects of *C. rotundus* kombucha on both in vitro and in vivo antioxidant activities. (**A**,**B**) The % inhibition of DPPH and ABTS free radicals in vitro, respectively. (**C**,**D**) the mean ± SD of % inhibition of DPPH and ABTS free radicals in *Drosophila* fed with a high-sugar diet (HSD), HSD supplemented with 10% *C. rotundus* kombucha (HSD + CRK10), HSD supplemented with 40% *C. rotundus* kombucha (HSD + CRK40), and HSD supplemented with 100% *C. rotundus* kombucha (HSD + CRK100), respectively. The One-Way ANOVA test was employed to determine differences among groups. * indicates statistically significant differences compared to the control group at *p* < 0.05.

**Figure 3 foods-12-04059-f003:**
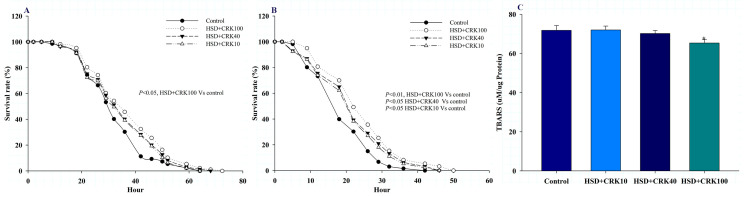
The effect of *C. rotundus* kombucha supplementation on hydrogen peroxide (**A**) and paraquat (**B**) resistance and lipid peroxidation (**C**) in *Drosophila* fed a high-sugar diet (HSD), HSD supplemented with 10% *C. rotundus* kombucha (HSD + CRK10), HSD supplemented with 40% *C. rotundus* kombucha (HSD + CRK40), and HSD supplemented with 100% *C. rotundus* kombucha (HSD + CRK100). Survival was analyzed using the Kaplan–Meier and log-rank tests. Differences were considered significant when the *p* value was less than 0.05. * indicates statistically significant differences compared to the HSD group at *p* < 0.05.

**Figure 4 foods-12-04059-f004:**
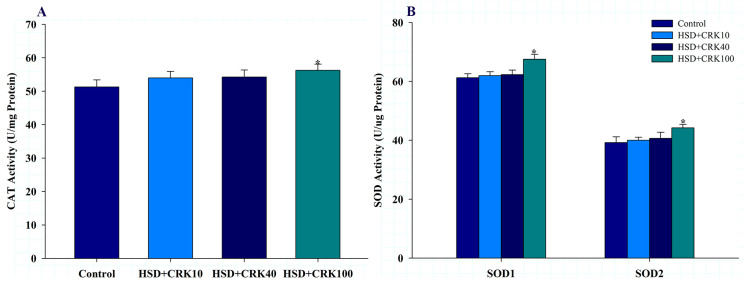
CAT (**A**) and SOD1 and SOD2 (**B**) levels in *Drosophila* fed a high-sugar diet (HSD), HSD supplemented with 10% *C. rotundus* kombucha (HSD + CRK10), HSD supplemented with 40% *C. rotundus* kombucha (HSD + CRK40), and HSD supplemented with 100% *C. rotundus* kombucha (HSD + CRK100). Data are presented as the mean ± SD. The One-Way ANOVA test was employed to determine differences among groups. * indicates statistically significant differences compared to the HSD group at *p* < 0.05.

**Figure 5 foods-12-04059-f005:**
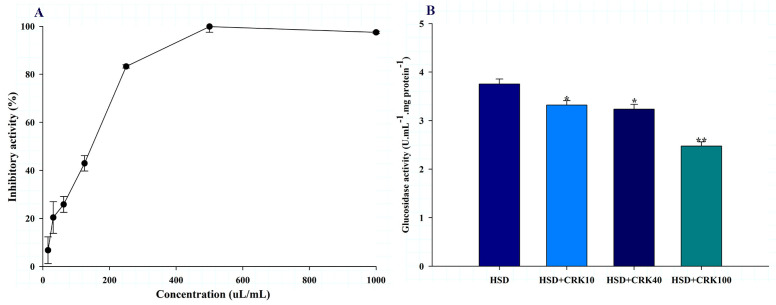
The effect of *C. rotundus* kombucha on the inhibition of α-glucosidase activity in vitro and in vivo. (**A**) The % inhibition of α-glucosidase activity in vitro. (**B**) α-glucosidase activity in *Drosophila* fed a high-sugar diet (HSD), HSD supplemented with 10% *C. rotundus* kombucha (HSD + CRK10), HSD supplemented with 40% *C. rotundus* kombucha (HSD + CRK40), and HSD supplemented with 100% *C. rotundus* kombucha (HSD + CRK100). The One-Way ANOVA test was employed to determine differences among groups. * and ** indicate statistically significant differences compared to the HSD group at *p* < 0.05 and *p* < 0.01, respectively.

**Table 1 foods-12-04059-t001:** The GC-MS profiling of *Cyperus rotundus* kombucha ferment.

No.	Name	Chemical Formula	RT	Previously Reported Biological Activity	Ref.
1	3,5-Dimethylpyrazole	C_5_H_8_N_2_	3.712	Hypoglycemic activity Anti-aging	[39,40]
2	Benzen-d5-amine	C_6_H_2_D_5_N	4.008	Not reported	NA
3	5-.alpha.-Aminoethyltetrazole	C_3_H_7_N_5_	5.156	Not reported	NA
4	2,4-Dihydroxy-2,5-dimethyl-3(2H)-furan-3-one	C_6_H_8_O_4_	6.112	Flavourant	[41]
5	Xylopyranoside, methyl-4-azido-4-deoxy-, .beta.-L	C_6_ H_11_N_3_O_4_	6.475	Not reported	NA
6	Propylamine, N,N,2,2-tetramethyl-, N-oxide	C_7_H_17_N O	6.992	Not reported	NA
7	Furaneol	C_6_H_8_O_3_	7.546	Not reported	NA
8	4H-Pyran-4-one, 2,3-dihydro-3,5-dihydroxy-6-methyl	C_6_H_8_O_4_	9.535	Antioxidant activity	[42]
9	Butyl-tert-butylisopropoxyborane	C_11_H_25_BO	10.119	Not reported	NA
10	1-Tetrazol-2-ylethanone	C_3_H_4_N_4_O	10.224	Not reported	NA
11	2-(4-Methyl-1H-1,2,3-triazol-1-yl)ethan-1-amine	C_5_H_10_N_4_	10.587	Antioxidant, anti-proliferative, and anti-atherosclerotic activities	[43]
12	Lactic acid, 2-methyl-, monoanhydride with 1-butaneboronic acid, cyclic ester	C_8_H_15_BO_3_	10.606	Not reported	NA
13	1-Tetrazol-2-ylethanone	C_3_H_4_N_4_O	10.903	Not reported	NA
14	Borinic acid, diethyl-	C_4_H_11_BO	10.684	Not reported	NA
15	5-Hydroxymethylfurfural	C_6_H_6_O_3_	11.294	Anticancer activity	[44]
16	10-Azido-1-decanethiol	C_10_H_21_N_3_S	12.604	Anti-fungal activity	[45]
17	.beta.-D-Glucosyloxyazoxymethane	C_8_H_16_N_2_O_7_	13.532	Not reported	NA
18	Ethanone, 1,1’-(2-ethyl-4,5-dimethyl-1,3,2-dioxaborolane-4,5-diyl)bis	C_10_H_17_BO_4_	14.144	Not reported	NA
19	1,3,2-Dioxaborolane, 2-ethyl-4-(3-oxiranylpropyl)-	C_9_H_17_BO_3_	15.330	Not reported	NA
20	Ethanone, 1-[5-(1,1-dimethylethyl)-2-ethyl-4-methyl-1,3,2-dioxaborolan-4-yl]-	C_11_H_21_BO_3_	18.036	Not reported	NA
21	3- Deoxy-D-mannoic lactone	C_6_H_10_O_5_	18.629	Antibacterial activity Antioxidant and alpha-glucosidase inhibitor activities	[46,47]
22	1-.beta.-d-Ribofuranosyl-3-[5-tetraazolyl]-1,2,4-triazole	C_8_H_11_N_7_O_4_	21.239	Analgesics, antipyretics, anti-convulsants, anti-inflammatory, immune modulatory activity	[48]
23	Nona-2,3-dienoic acid, ethyl ester	C_11_H_18_O_2_	24.949	Antiviral activity	[49]
24	Stigmasterol	C_29_H_48_O	29.530	Anti-osteoarthritic activity Anti-hypercholestrolemic activity Anti-tumor Antioxidant Antimutagenic activity Anti-inflammatory activity CNS activities	[50,51,52,53,54,55,56,57,58]

NA = Not available.

## Data Availability

The data presented in this study are available on request from the corresponding author.

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
