# Peer review of "In Vitro and In Vivo Evaluation for Antioxidant and Anti-Diabetic Properties of *Cyperus rotundus* L. Kombucha"

_foods, 2023, doi:10.3390/foods12224059_

Round 1
Reviewer 1 Report
Comments and Suggestions for Authors
The manuscript evaluates the in vitro and in vivo antioxidant and anti-diabetic properties of Cyperus rotundus L. kombucha. The topic is interesting; However, the manuscript has several problems:
1. L 1; please define the type of the paper.
2. L 27; "SOD2" not "SOD2".
3. L 28-30; express the reduction rate.
4. L 57-69; this paragraph needs some references.
5. Scientific names should be presented in italics. For example, L 86, 113, etc. (check the entire manuscript)
6. Check the format of citations. For example, L 175 should be "Pistia-Brueggeman and Hollingsworth [34]". Check the entire manuscript.
7. L 205; CO2.
8. L 329; this figure should be "Figure 1". Also, provide a figure with a higher resolution.
9. Please add the significant letters to the chart bars (Figures 2 to 5) ("a" should be utilized for the highest value). In addition, the treatments should present in the following order: HSD, HSD+CRK10, HSD+CRK40, and HSD+CRK100 (also check part 2.7).
10. After the discussion part, add the "conclusion" section as a separate part.
Author Response
Reviewer #1
Q1. L 1; please define the type of the paper.
Answer The type of the paper was defined as an article, as suggested by the reviewer.
Q2. L 27; "SOD2" not "SOD2".
Answer We made the changes as suggested by the reviewer.
Q3. L 28-30; express the reduction rate.
Answer The rate of reduction has been added in lines 28-30, as suggested by the reviewer.
Q4. L 57-69; this paragraph needs some references.
Answer We have added three references as follows:
[27] Hossain, U., Das, A. K., Ghosh, S., Sil, P.C. An overview on the role of bioactive α-glucosidase inhibitors in ameliorating diabetic complications. Food Chem. Toxicol. 2020, 145, 111738.
[28] Ghani, U. Re-exploring promising α-glucosidase inhibitors for potential development into oral anti-diabetic drugs: Finding needle in the haystack. Eur. J. Med. Chem. 2015, 103, 133–162.
[29] Apovian, C. M., Okemah, J., O'Neil, P. M. Body weight considerations in the management of type 2 diabetes. Adv. Ther. 2019, 36, 44–58.
Q5. Scientific names should be presented in italics. For example, L 86, 113, etc. (check the entire manuscript)
Answer Thank you the reviewer for the comment. We have carefully checked the entire manuscript and corrected all scientific names.
Q6. Check the format of citations. For example, L 175 should be "Pistia-Brueggeman and Hollingsworth [34]". Check the entire manuscript.
Answer We have edited and checked the entire manuscript as suggested by reviewer.
Q7. L 205; CO2.
Answer We have edited as suggested by reviewer.
Q8. L 329; this figure should be "Figure 1". Also, provide a figure with a higher resolution.
Answer We have edited as suggested by reviewer.
Q9. Please add the significant letters to the chart bars (Figures 2 to 5) ("a" should be utilized for the highest value). In addition, the treatments should present in the following order: HSD, HSD+CRK10, HSD+CRK40, and HSD+CRK100 (also check part 2.7).
Answer Figure 2 to 5, we have edited the order of treatments as suggested by reviewer.
Q10. After the discussion part, add the "conclusion" section as a separate part.
Answer We added the conclusion section as recommended by the reviewer.
Reviewer 2 Report
Comments and Suggestions for Authors
Comments on the manuscript marked in the attached file

Author Response
Reviewer #2
2.2 Plant materials and preparation of kombucha
Q1. When and on what date the collection was made.
Answer The sample collection date was added as suggested by the reviewer (L 112-113).
Q2. At what temperature was it filtered?
Answer A solution was boiled and then filtered at 95°C. We have included the filtered temperature in the manuscript (L 117).
Q3. What did the leavening agent contain?
Answer In this study, we added a 10% SCOBY into the boiled solution to make kombucha. Therefore, we removed the term 'leavening agent' from the manuscript to avoid confusion with leavening agents used in baking, such as yeast or baking powder (L119-120).
2.4 HPLC analysis of C. rotundus kombucha
Q4. Describe gradient flows
Answer The HPLC conditions were included in the manuscript (L 139-142).
3.1 GC-MS chemical profiles of C. rotundus kombucha
Q5. These are not chromatograms but identified compounds, i.e. compound profiles.
Answer Thank you, reviewer, for the for the comment. We have edited as suggested (L 310).
Q6. The content of phenols and flavonoids is shown in Figure 1. The wording is incorrect because only chlorogenic acid and epicatechin have been identified. These are only representatives of phenolic acids and flavonols. The text is signed as Figure 2, or rather, according to the text, it should be Figure No. 1. There are other peaks in the chromatogram that have not been identified. Therefore, the wording of the sentence should be changed.
Answer We changed the wording from 'phenols and flavonoids' to 'phytochemical compounds' (L 316). Additionally, the figure reference was corrected to 'Figure 1' both in the text and in the figure caption (L320, L328).
Figure 3
Q7. Explain the meaning of one and two stars on the chart.
Answer As suggested by the reviewer, we have explained the significance of the one and two stars on the chart.
Reviewer 3 Report
Comments and Suggestions for Authors
In this study, the antioxidant capacity and glucosidase inhibition of a beverage prepared from Cyperus rotundus L. are investigated. The study is conducted both in vitro and in vivo, using Drosophila as a model. The most likely weak aspect of the study is the phytochemical characterization, which is primarily qualitative in nature.
Section 3.1 GC-MS chemical profiles of C. rotundus kombucha, includes also the HPLC characterization of phenolics. It is suggested to change the title of the section to simply “Chemical profiles of C. rotundus kombucha ".
Section 3.1: Please explain how did you identified the different compounds in GC-MS.
Line 312: Please explain how did you quantified the phenolics. Did you use calibration with standars? If that is the case, you should include the range of concentrations used and the calibration curve equation.
Figure 2C: * is missing for columns HSD+CRK40 and HSD+CRK10 (according with the text line 345-347 the values are statistically different from the control group).
Author Response
Reviewer #3
Q1. Section 3.1 GC-MS chemical profiles of C. rotundus kombucha, includes also the HPLC characterization of phenolics. It is suggested to change the title of the section to simply “Chemical profiles of C. rotundus kombucha".
Answer Thank you for the suggestion. We have made the changes as the reviewer suggested (L 309).
Q2. Section 3.1: Please explain how did you identified the different compounds in GC-MS.
Answer We identified the different compounds in GC-MS by using NIST Mass Spectral Database. We have incorporated these details into Section 2.3 of the manuscript (L 132-133).
Q3. Line 312: Please explain how did you quantified the phenolics. Did you use calibration with standard? If that is the case, you should include the range of concentrations used and the calibration curve equation.
Answer We quantified the amounts of chlorogenic acid and epicatechin using a standard curve. The range for chlorogenic acid was 3.13-50.00 µg/mL, described by the equation Y = 273991x + 788,198. The range for epicatechin was 1.56-50.00 µg/mL, given by the equation Y = 166769x + 560043. We have incorporated these details into the manuscript (L 317-320).
Q4. Figure 2C: * is missing for columns HSD+CRK40 and HSD+CRK10 (according with the text line 345-347 the values are statistically different from the control group).
Answer Thank you for the suggestion. We have added * to the columns as the reviewer suggested.
Round 2
Reviewer 3 Report
Comments and Suggestions for Authors
All the observations made by the reviewer have been appropriately addressed.